# CH$_3$SH and H$_2$S Sensing Properties of V$_2$O$_5$/WO$_3$/TiO$_2$ Gas Sensor

Takafumi Akamatsu *, Toshio Itoh, Akihiro Tsuruta and Yoshitake Masuda

National Institute of Advanced Industrial Science and Technology (AIST), Innovative Functional Materials Research Institute, 2266-98, Anagahora, Shimo-Shidami, Nagoya-shi 463-8560, Japan; itoh-toshio@aist.go.jp (T.I.); a.tsuruta@aist.go.jp (A.T.); masuda-y@aist.go.jp (Y.M.)

* Correspondence: t-akamatsu@aist.go.jp; Tel.: +81-52-736-7602

**Abstract:** Resistive-type semiconductor-based gas sensors were fabricated for the detection of methyl mercaptan and hydrogen sulfide. To fabricate these sensors, V$_2$O$_5$/WO$_3$/TiO$_2$ (VWT) particles were deposited on interdigitated Pt electrodes. The vanadium oxide content of the utilized VWT was 1.5, 3, or 10 wt.%. The structural properties of the VWT particles were investigated by X-ray diffraction and scanning electron microscopy analyses. The resistance of the VWT gas sensor decreased with increasing methyl mercaptan and hydrogen sulfide gas concentrations in the range of 50 to 500 ppb. The VWT gas sensor with 3 wt.% vanadium oxide showed high methyl mercaptan and hydrogen sulfide responses and good gas selectivity against hydrogen at 300 °C.

**Keywords:** gas sensor; vanadia–tungsten–titania; semiconductor; methyl mercaptan; hydrogen sulfide





## 1. Introduction

Human breath contains small amounts of biomarker gases, such as hydrogen (H$_2$), nitric oxide (NO), carbon monoxide (CO), methane (CH$_4$), and various volatile organic components, at low concentrations ranging from several parts per billion (ppb) to parts per million (ppm) [1–4]. The organoleptic intensity of oral malodor correlates with the levels of methyl mercaptan (CH$_3$SH) and hydrogen sulfide (H$_2$S), which are volatile sulfur compounds, in human breath [4]. Although monitoring CH$_3$SH and H$_2$S concentrations in human breath is one of the best noninvasive screening tests for early diagnosis, the screening process requires an analytical device with sufficient accuracy at the ppb level. The concentration level of H$_2$ in human breath has been measured to be several tens of ppm, which is higher than those of the other gases [5].

Semiconductor-based gas sensors are effective for respiratory gas analysis from the standpoints of cost, compactness, and power consumption [6–11]. Su et al. reported that for NO$_2$ sensing, a wearable, alveolus-inspired active membrane sensor based on the WO$_3$ system showed excellent selectivity and thermal and humidity stability [6]. Jha et al. prepared chemiresistive sensors using MoSe$_2$ nanoflakes for ppb-level H$_2$S gas detection [7]. Li et al. synthesized CuO nanosheets by a hydrothermal method and used them to fabricate a sensor, which showed ppb-level H$_2$S gas response [8].

Our group has reported a resistive-type SO$_2$ gas sensor based on V$_2$O$_5$/WO$_3$/TiO$_2$ (VWT) [12,13]. Although this gas sensor has been confirmed to respond well to 20 to 5000 ppm SO$_2$ gas, its responsiveness to methyl mercaptan and hydrogen sulfide has not been investigated. In this study, VWT-based gas sensors with V$_2$O$_5$ contents of 1.5, 3.0, and 10 wt.% were prepared. The responsiveness of the gas sensor to ppb-level methyl mercaptan and hydrogen sulfide was investigated. To discuss the gas selectivity of the gas sensor, the sensor response to H$_2$ gas at the ppm level was investigated.

## 2. Materials and Methods

After stirring 10 g of $TiO_2$ nanoparticles (Evonik Aeroxide P 25) in 115 mL of ion-exchanged water, 0.444 g of ammonium vanadate (FUJIFILM Wako Pure Chemical Corporation, Osaka, Japan), 1.27 g of ammonium paratungstate (Sigma-Aldrich, St. Louis, MO, USA), and 12 mL of 4% methylamine aqueous solution were added. After mixing for 24 h, the solution was dried at 85 °C to obtain a dry sample. The dry sample was ground and calcined at 450 °C for 5 h to obtain $3V_2O_5/10WO_3/87TiO_2$ (wt.%) (3 VWT) particles. $1.5V_2O_5/10WO_3/88.5TiO_2$ (1.5 VWT) and $10V_2O_5/10WO_3/80TiO_2$ (10 VWT) particles were prepared in the same manner.

The sample powders were characterized by X-ray diffraction (XRD) and scanning electron microscopy (SEM) analyses. XRD analysis was carried out using a SmartLab X-ray diffractometer (Rigaku Corporation, Tokyo, Japan) equipped with a copper source (Cu K$\alpha$) and one-dimensional high-speed detector (D/teX Ultra 250). The X-ray generator was operated at 40 kV and 30 mA. SEM analysis was performed using a JSM-6335FM microscope (JEOL Ltd., Tokyo, Japan) equipped with a field-emission gun. To obtain the Brunauer–Emmett–Teller (BET) surface areas, the nitrogen adsorption isotherms of sample particles (degassed at 300 °C for 2 h) were measured at 77 K in liquid nitrogen using a NOVA 4200e surface area and pore size analyzer (Quantachrome Instruments, Boynton Beach, FL, USA).

A ceramic paste of VWT particles was prepared by mixing an organic vehicle consisting of 10 wt.% ethyl cellulose and 90 wt.% terpineol. The mixing weight ratio of VWT and the organic vehicle was 2:1.33. The paste was screen-printed onto an alumina substrate with Pt interdigital electrodes with line and space definitions of 100 μm each. These sensor elements were baked at 750 °C for 20 min to prepare a gas sensor with a sensing film having a thickness of approximately 5 μm. Figure 1 shows a schematic illustration of the sensor.

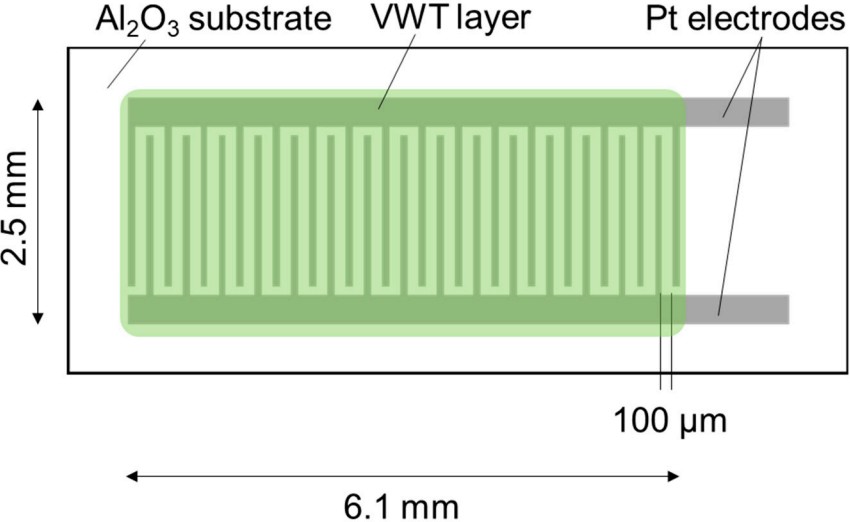

**Figure 1.** Schematic illustration of the $V_2O_5/WO_3/TiO_2$ (VWT) sensor.

The gas responsiveness of the sensor was examined using a test chamber in an electric furnace at 300–500 °C. After placing the sensor in the chamber, dry air was flowed into the chamber for 1 h, followed by the sample gas in dry air for 1 h, at a flow rate of 200 mL/min. The $CH_3SH$ and $H_2S$ gas concentrations were controlled to 0, 50, 250, and 500 ppb in dry air. Sensor resistances in various gas atmospheres were measured by the two-terminal method with a K2700 digital multimeter (Keithley) at 5 s intervals. The response value ($S$) was defined by the following equation: $S = R_a/R_g$, where $R_a$ denotes the resistance in dry air before sample gas exposure and $R_g$ is the resistance after 1 h of sample gas exposure.

The response of the sensors to $H_2$ gas was investigated using the same apparatus as that used to measure the $CH_3SH$ and $H_2S$ gas responses. The $H_2$ gas concentration was controlled to 0, 250, and 500 ppm in dry air.

## 3. Results and Discussion

Figure 2 shows the XRD patterns of the 1.5 VWT, 3 VWT, and 10 VWT particles. Peaks of $TiO_2$ (anatase and rutile) and $WO_3$ were observed in all XRD patterns. In addition, peaks of $V_2O_5$ were confirmed in the 10 VWT XRD pattern. No $V_2O_5$ peaks were observed in the XRD patterns of 1.5 VWT and 3 VWT owing to the lower $V_2O_5$ content. Since the peaks assigned to $TiO_2$, $WO_3$, and $V_2O_5$ were observed, no structural changes due to the formation of the complex oxide of V-W-Ti-O were observed.

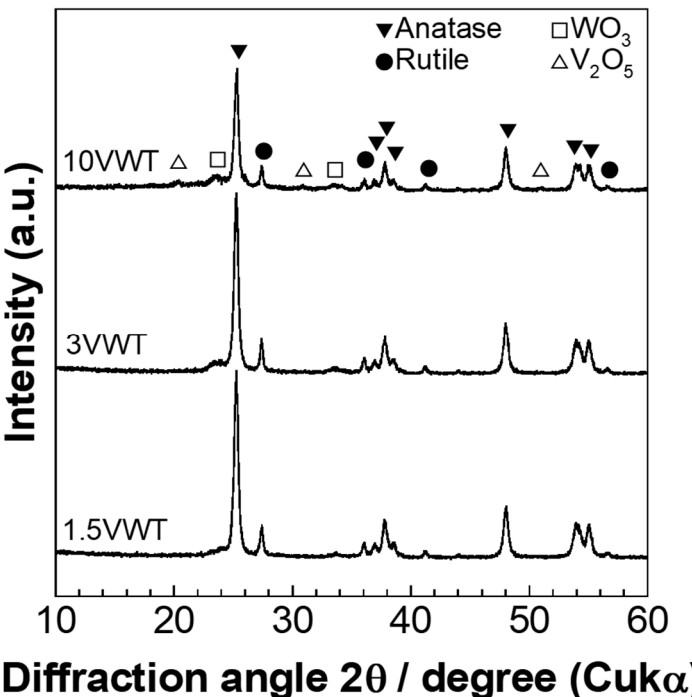

**Figure 2.** XRD patterns of 1.5 VWT, 3 VWT, and 10 VWT particles.

Figure 3 shows the SEM images of the 1.5 VWT, 3 VWT, and 10 VWT particles. For all samples, uniform particles with diameters of several tens of nanometers were confirmed. As these diameters are almost the same as those of $TiO_2$ particles (P 25), it is assumed that $V_2O_5$ and $WO_3$ are supported on the surface of the $TiO_2$ particles in VWT. The BET surface areas of the 1.5 VWT, 3 VWT, and 10 VWT particles were 46, 47, and 34 $m^2/g$, respectively. When compared, 10 VWT had a slightly lower BET surface area than 1.5 VWT and 3 VWT. Since the specific surface area of P 25 $TiO_2$ is 57.4 $m^2/g$ [14], decreasing the amount of $TiO_2$ decreased the BET surface area of VWT. Although the difference in particle morphology could not be confirmed in Figure 3, the difference in the specific surface area may be due to the difference in particle morphology.

Figure 4 shows the Arrhenius plots of $R_a$ for the 1.5 VWT, 3 VWT, and 10 VWT gas sensors. $R_a$ decreased with decreasing $TiO_2$ content in VWT. $R_a$ changed exponentially with the reciprocal of temperature between 300 and 500 °C, as shown in the following equation: $R_a = R_0 \exp(-E/RT)$, where $E$ is the activation energy, $T$ is the temperature in Kelvin, $R$ is the gas constant, and $R_0$ is the pre-exponential factor. This indicates that there is no change in the mechanism of electron conduction in VWT between 300 and 500 °C. The activation energies for electron conduction in the 1.5 VWT, 3 VWT, and 10 VWT gas sensors were 0.57, 0.40, and 0.34 eV, respectively. It is considered that the sensor resistance decreased owing to the decrease in the $TiO_2$ content in VWT, resulting in a decrease in the activation energy.

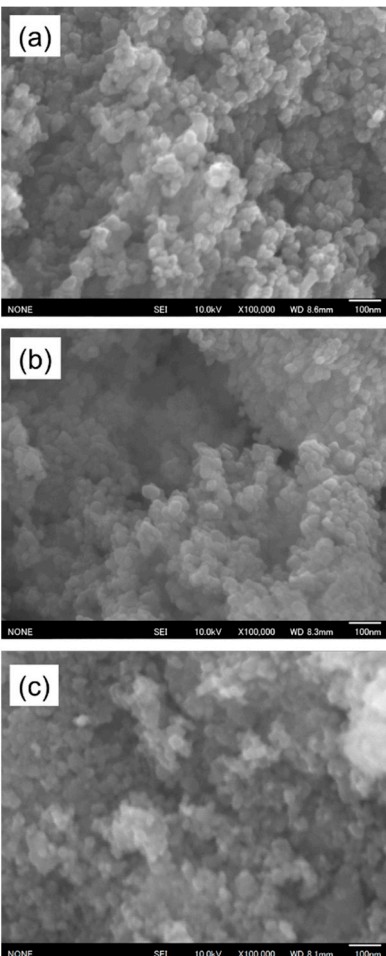

**Figure 3.** SEM photographs of (**a**) 1.5 VWT, (**b**) 3 VWT, and (**c**) 10 VWT particles.

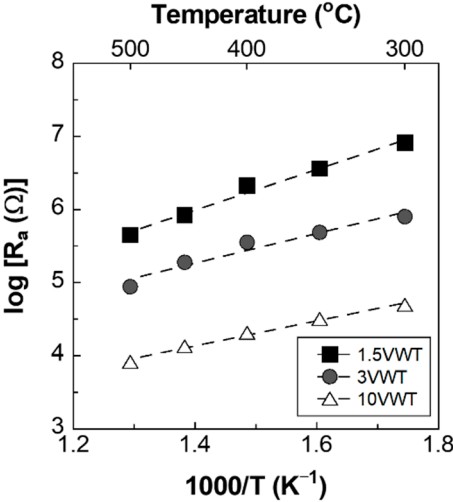

**Figure 4.** Arrhenius plots of the resistance in dry air ($R_a$) of the 1.5 VWT, 3 VWT, and 10 VWT gas sensors.

Figure 5 shows the $CH_3SH$ gas (50, 250, and 500 ppb) response characteristics of the 1.5 VWT, 3 VWT, and 10 VWT gas sensors at 300 °C. The resistance of all sensors began to decrease with $CH_3SH$ exposure. This is the response of a typical *n*-type semiconductor gas sensor. All gas sensors showed a clear response, even at 50 ppb $CH_3SH$. The resistance of

all sensors decreased with increasing $CH_3SH$ gas concentration. The response of the *n*-type semiconductor gas sensor to $H_2S$ gas was similar.

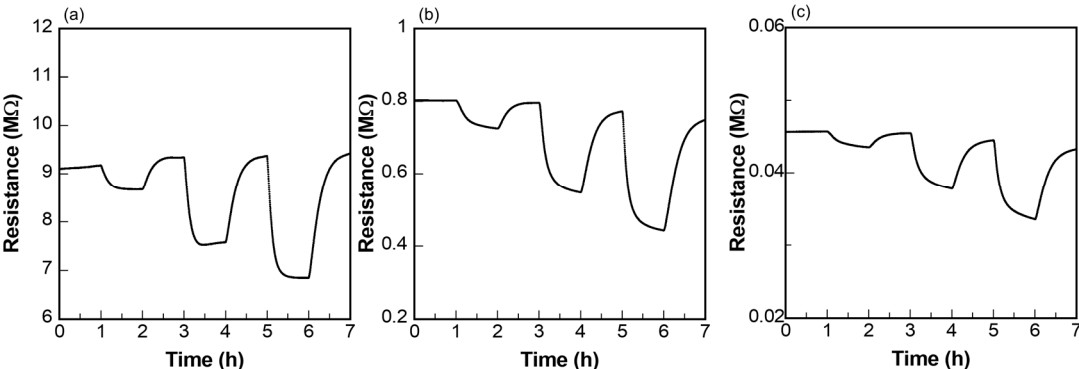

**Figure 5.** $CH_3SH$ gas (50, 250, and 500 ppb) response characteristics of the (**a**) 1.5 VWT, (**b**) 3 VWT, and (**c**) 10 VWT gas sensors at 300 °C.

Figure 6 shows the relationship between the response value to 500 ppb $CH_3SH$ and 500 ppb $H_2S$ and operating temperature. High $CH_3SH$ response values ($S$ = 1.74, 1.74, and 1.33) were obtained for the 1.5 VWT, 3 VWT, and 10 VWT gas sensors at 400, 300, and 300 °C, respectively. In addition, high $H_2S$ response values ($S$ = 1.29, 1.44, and 1.13) were obtained for the 1.5 VWT, 3 VWT, and 10 VWT gas sensors at 400, 400, and 300 °C, respectively. The response to $CH_3SH$ was higher than that to $H_2S$. The reaction mechanism between the $CH_3SH$ and $H_2S$ gases and the oxygen ($O_2^{2-}$) adsorbed on the surface of VWT is expected to be as follows:

$$CH_3SH + 3O_2^{2-} \rightarrow CO_2 + 2H_2O + SO_2 + 6e^- \tag{1}$$

$$H_2S + 3/2O_2^{2-} \rightarrow H_2O + SO_2 + 3e^- \tag{2}$$

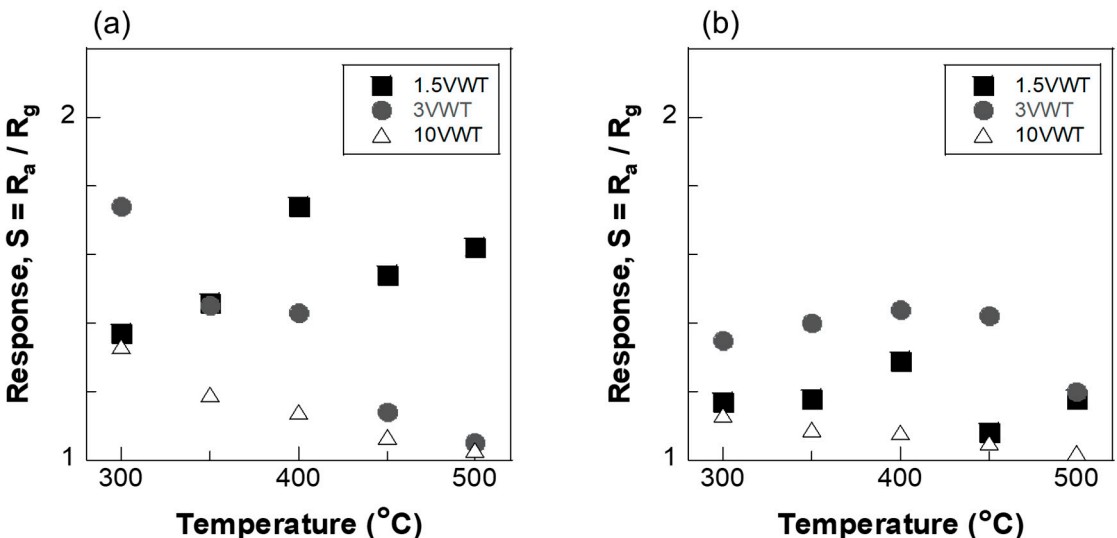

**Figure 6.** Relationship between the response values ($S$) to (**a**) 500 ppb $CH_3SH$ and (**b**) 500 ppb $H_2S$ and operating temperature. $S$ was calculated as the ratio of the resistance in dry air before ($R_a$) and after ($R_g$) sample gas exposure.

Equation (1) shows that 1 mol of $CH_3SH$ reacts with the adsorbed oxygen to form 6 mol of conduction electrons ($e^-$). Equation (2) shows that 1 mol of $H_2S$ reacts with the adsorbed oxygen to form 3 mol of $e^-$. The generated $e^-$ moves to VWT and reduces the sensor resistance. Therefore, if the reaction rates of $CH_3SH$ and $H_2S$ with the adsorbed oxygen are almost the same, it is considered that the response to $CH_3SH$ is higher than that

to $H_2S$. The 10 VWT gas sensor showed smaller response values to the $CH_3SH$ and $H_2S$ gases. As $V_2O_5$ oxidizes $SO_2$ to $SO_3$, it is thought that the oxidation of the $SO_2$ generated from $CH_3SH$ and $H_2S$ leads to a decreased sensor response [15].

Figure 7 shows the relationship between the response value and $CH_3SH$, $H_2S$, and $H_2$ gas concentrations at 300 and 400 °C. At 300 °C, the $CH_3SH$ response value of the 3 VWT gas sensor was large. At 400 °C, the $CH_3SH$ response value of the 1.5 VWT gas sensor was also large. The 1.5 VWT gas sensor showed a slight response to $H_2$ gas with $S = 1.1$–1.22. On the other hand, the 3 VWT and 10 VWT gas sensors showed no response to $H_2$ gas with $S = 1.00$ and $S = 1.02$, respectively. Since the $H_2$ gas response decreased with increasing $V_2O_5$ content in VWT, $V_2O_5$ seems to suppress the response to $H_2$ gas. We will investigate the mechanism of suppressing the $H_2$ gas response by $V_2O_5$ in the future.

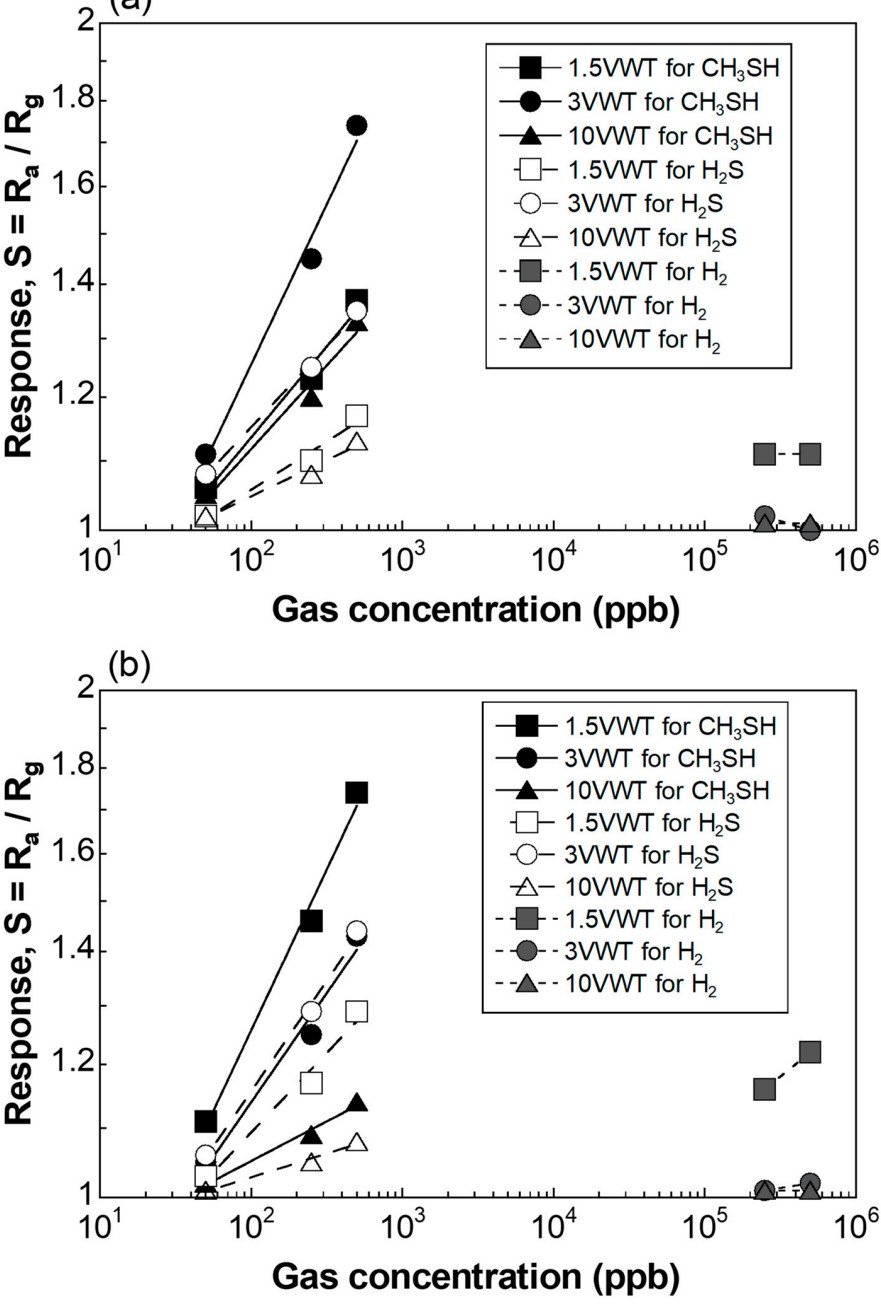

**Figure 7.** Relationship between the response value $S$ and $CH_3SH$, $H_2S$, and $H_2$ gas concentrations at (**a**) 300 and (**b**) 400 °C. $S$ was calculated as the ratio of the resistance in dry air before ($R_a$) and after ($R_g$) sample gas exposure.

It was found that at 300 °C, the 3 VWT gas sensor had good gas selectivity against $H_2$ gas and a high $CH_3SH$ response of 500 ppb ($S$ = 1.74). This $CH_3SH$ response value is smaller than that of the $WO_3$ gas sensor in earlier reports [10,11]. In this study, the $H_2S$, $CH_3SH$, and $H_2$ gas response characteristics in dry atmosphere were measured only once. The long-term stability, reproducibility, and influence of humidity will be investigated in a future study. Although it is necessary to improve the $CH_3SH$ response value, it is considered that the 3 VWT gas sensor has a high potential as a gas sensor for the detection of halitosis because it has excellent gas selectivity against $H_2$ gas.

## 4. Conclusions

In this study, $CH_3SH$ and $H_2S$ gas sensors using VWT as a sensitive film were prepared, and their sensing characteristics were investigated. The synthesized VWT particles had a diameter of several tens of nanometers, and no V-W-Ti-O composite oxide was formed. The $CH_3SH$ response value of the VWT gas sensor was higher than the $H_2S$ response value. The 3 VWT gas sensor showed high $CH_3SH$ and $H_2S$ gas responses and good gas selectivity against $H_2$ gas.

**Author Contributions:** Conceptualization, T.A., T.I., A.T. and Y.M.; methodology, T.A.; validation, T.A.; formal analysis, T.A., T.I., A.T. and Y.M.; investigation, T.A.; resources, T.A., T.I., A.T. and Y.M.; data curation, T.A., T.I. and A.T.; writing—original draft preparation, T.A.; writing—review and editing, T.A.; visualization, T.A.; supervision, T.A. and Y.M.; project administration, T.A. and Y.M. All authors have read and agreed to the published version of the manuscript.

**Funding:** This research received no external funding.

**Institutional Review Board Statement:** Not applicable.

**Informed Consent Statement:** Not applicable.

**Data Availability Statement:** Not applicable.

**Acknowledgments:** We express our sincere thanks to Ayako Uozumi of AIST for her valuable support in the BET analysis.

**Conflicts of Interest:** The authors declare no conflict of interest.

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
