# Peer review of "CH3SH and H2S Sensing Properties of V2O5/WO3/TiO2 Gas Sensor"

_chemosensors, doi:10.3390/chemosensors9050113_

Round 1

Reviewer 1 Report

The manuscript entitled “CH3SH and H2S sensing properties of V2O5/WO3/TiO2 gas sensor” reported a resistive-type semiconductor-based gas sensor for the methyl mercaptan and hydrogen sulfide detection in the range of 50 to 500 ppb. The selectivity was tested against hydrogen at 300 °C. The manuscript has the following issues to be addressed.

  1. The interdigital electrode pattern was not demonstrated in the manuscript. It is recommended that the authors can add a figure to show the pattern of the electrodes.
  2. In Figure 3, the temperatures were only varied in the range of 300-500 °C, however, the results showed that the optimal response was at 300 ° It deserves to investigate the sensing performance at 250 °C to show that the 300 °C was actually the best operating temperature of proposed gas sensors.
  3. How was the repeatability of the fabricated sensor towards methyl mercaptan and hydrogen sulfide?
  4. How stable the fabricated sensor was? Is it able to be reused after a single test? How the recovery of the sensing single looks like?
  5. It would be good to add acetone in the selectivity test investigations if the experimental set-up allows.
  6. What was the testing humidity condition when operating the experiment? Is the fabricated sensor resistant to the moisture?

Author Response

Journal: Chemosensors

Manuscript ID: chemosensors-1177377

Authors: Takafumi Akamatsu, Toshio Itoh, Akihiro Tsuruta, Yoshitake Masuda

Thank you for carefully reviewing our manuscript. Regarding the concerns you have raised about this work, I would like to request your kind understanding and reevaluation based on the following points.

Reviewer’s comments:

Reviewer #1: The manuscript entitled “CH3SH and H2S sensing properties of V2O5/WO3/TiO2 gas sensor” reported a resistive-type semiconductor-based gas sensor for the methyl mercaptan and hydrogen sulfide detection in the range of 50 to 500 ppb. The selectivity was tested against hydrogen at 300 °C. The manuscript has the following issues to be addressed.

  1. The interdigital electrode pattern was not demonstrated in the manuscript. It is recommended that the authors can add a figure to show the pattern of the electrodes.

(Reply)

We have added a figure (Figure 1 in the revised manuscript) of the interdigital electrode pattern.

(revised manuscript  p. 2, line 70)

  1. In Figure 3, the temperatures were only varied in the range of 300-500 °C, however, the results showed that the optimal response was at 300 ° It deserves to investigate the sensing performance at 250 °C to show that the 300 °C was actually the best operating temperature of proposed gas sensors.

(Reply)

              As the reviewer noted, the sensing performance should also be tested at the lower temperature range (e.g., 250°C) to unequivocally show that 300°C is the optimum operating temperature.. However, we currently do not have H2S and CH3SH gas cylinders. We will confirm the sensor response after they are obtained. Thank you for your advice.

  1. How was the repeatability of the fabricated sensor towards methyl mercaptan and hydrogen sulfide?

(Reply)

              In this study, the H2S, CH3SH, and H2 gas response characteristics were measured only once. We will investigate the repeatability in a future study. To clarify this, a sentence has been added as follows.

(revised manuscript  p. 6, line 156)

              … [10,11]. In this study, the H2S, CH3SH, and H2 gas response characteristics in dry atmosphere were measured only once. The long-term stability, reproducibility, and influence of humidity will be investigated in a future study. Although …

  1. How stable the fabricated sensor was? Is it able to be reused after a single test? How the recovery of the sensing single looks like?

(Reply)

              In this study, the H2S, CH3SH, and H2 gas response characteristics were measured only once. We will investigate the long-term stability in a future study. To clarify this, a sentence has been added as follows.

(revised manuscript  p. 6, line 156)

              … [10,11]. In this study, the H2S, CH3SH, and H2 gas response characteristics in dry atmosphere were measured only once. The long-term stability, reproducibility, and influence of humidity will be investigated in a future study. Although …

  1. It would be good to add acetone in the selectivity test investigations if the experimental set-up allows.

(Reply)

              Currently, we do not have an acetone gas cylinder. We will confirm the sensor response after it is obtained.

  1. What was the testing humidity condition when operating the experiment? Is the fabricated sensor resistant to the moisture?

(Reply)

              In this study, the H2S, CH3SH, and H2 gas response characteristics in dry atmosphere were measured only once. We will investigate the testing humidity condition in a future study. To clarify this, a sentence has been added as follows.

(revised manuscript  p. 6, line 156)

              … [10,11]. In this study, the H2S, CH3SH, and H2 gas response characteristics in dry atmosphere were measured only once. The long-term stability, reproducibility, and influence of humidity will be investigated in a future study. Although …

(Over)

Reviewer 2 Report

In this paper, the authors developed a Resistive-type semiconductor-based gas sensors for the detection of methyl mercaptan and hydrogen sulfide. To fabricate these sensors, V2O5/WO3/TiO2 (VWT) particles were deposited on interdigitated Pt electrodes,The CH3SH response value of the VWT gas sensor was higher than the H2S re-sponse value. The 3VWT gas sensor showed high CH3SH and H2S gas responses and good gas selectivity against H2 gas. However, I cannot recommend accept this article at current version unless the authors address the following concerns.

1.In page 2, the author mentioned that ” alternately flowed into the chamber”. What does it mean? How to define the word “alternately”?

  1. In figure 1、3、6,The ruler of the chart looks too big
  2. BET test should be added to match with the different morphologies in Fig. 2
  3. Long-term stability should be given.
  4. How about the reproducibility and selectivity of the prepared sensor?
  5. The author investigated on the nanomaterials based gas sensor. Some relative papers may enrich and strengthen the concept and background of this work as the references. ACS Nano, 2020, 14, 6067-6075; Adv. Funct. Mater., 2021, 2010962; Sensors and Actuators B: Chemical, 2017, 247,540-545; Sensors and Actuators B: Chemical, 2019, 297,126687; Applied Surface Science, 2017, 423,492-497;Applied Surface Science, 2017, 445,89-96.

Author Response

Journal: Chemosensors

Manuscript ID: chemosensors-1177377

Authors: Takafumi Akamatsu, Toshio Itoh, Akihiro Tsuruta, Yoshitake Masuda

Thank you for carefully reviewing our manuscript. Regarding the concerns you have raised about this work, I would like to request your kind understanding and reevaluation based on the following points.

Reviewer’s comments:

Reviewer #2: In this paper, the authors developed a Resistive-type semiconductor-based gas sensors for the detection of methyl mercaptan and hydrogen sulfide. To fabricate these sensors, V2O5/WO3/TiO2 (VWT) particles were deposited on interdigitated Pt electrodesThe CH3SH response value of the VWT gas sensor was higher than the H2S response value. The 3VWT gas sensor showed high CH3SH and H2S gas responses and good gas selectivity against H2 gas. However, I cannot recommend accept this article at current version unless the authors address the following concerns.

  1. In page 2, the author mentioned that ” alternately flowed into the chamber”. What does it mean? How to define the word “alternately”?

(Reply)

As the reviewer noted, the term “alternately” may be unclear. Hence, the sentence has been revised as follows.

(revised manuscript  p. 2, line 73)

              … the chamber, dry air was flowed into the chamber for 1 h, followed by the sample gas in dry air for 1 h, at a …

  1. In figure 136The ruler of the chart looks too big

(Reply)

              The axis size in Figures 1, 3, and 6 (Figures 2, 4, and 7, respectively, in the revised manuscript) has been reduced.

  1. BET test should be added to match with the different morphologies in Fig. 2

(Reply)

              We have performed N2 adsorption experiments to determine the BET surface areas. We have added sentences describing the procedure and results in the revised manuscript.

(revised manuscript  p. 2, line 59)

              … gun. To obtain the Brunauer–Emmett–Teller (BET) surface areas, the nitrogen adsorption isotherms of sample particles (degassed at 300°C for 2 h) were measured at 77 K in liquid nitrogen using a NOVA 4200e surface area and pore analyzer (Quantachrome Instruments).

(revised manuscript  p. 3, line 93)

              … in VWT. The BET surface areas of the 1.5VWT, 3VWT, and 10VWT particles were 46, 47, and 34 m2/g, respectively. 10VWT had a slightly lower BET surface area than 1.5VWT and 3VWT. Since the specific surface area of P 25 TiO2 is 57.4 m2/g [14], decreasing the amount of TiO2 decreased the BET surface area of VWT. Although the difference in particle morphology could not be confirmed in Fig. 3, the difference in the specific surface area may be due to the difference in particle morphology.

(revised manuscript  p. 8, line 211)

  1. Suttiponparnit, K.; Jiang, J.; Sahu, M.; Suvachittanont, S.; Charinpanitkul, T.; Biswas, P. Role of Surface Area, Primary Particle Size, and Crystal Phase on Titanium Dioxide Nanoparticle Dispersion Properties. Nanoscale Res. Lett. 2011, 6, 27.

  1. Long-term stability should be given.

(Reply)

              In this study, the H2S, CH3SH, and H2 gas response characteristics were measured only once. We will investigate the long-term stability in a future study. To clarify this, a sentence has been added as follows.

(revised manuscript  p. 6, line 156)

              … [10,11]. In this study, the H2S, CH3SH, and H2 gas response characteristics in dry atmosphere were measured only once. The long-term stability, reproducibility, and influence of humidity will be investigated in a future study. Although …

  1. How about the reproducibility and selectivity of the prepared sensor?

(Reply)

In this study, we investigated the selectivity against H2 gas (see Figure 7 in the revised manuscript). However, the H2S, CH3SH, and H2 gas response characteristics were measured only once. We will investigate the reproducibility in a future study. To clarify this, a sentence has been added as follows.

(revised manuscript  p. 6, line 156)

              … [10,11]. In this study, the H2S, CH3SH, and H2 gas response characteristics in dry atmosphere were measured only once. The long-term stability, reproducibility, and influence of humidity will be investigated in a future study. Although …

  1. The author investigated on the nanomaterials based gas sensor. Some relative papers may enrich and strengthen the concept and background of this work as the references. ACS Nano, 2020, 14, 6067-6075; Adv. Funct. Mater., 2021, 2010962; Sensors and Actuators B: Chemical, 2017, 247,540-545; Sensors and Actuators B: Chemical, 2019, 297,126687; Applied Surface Science, 2017, 423,492-497; Applied Surface Science, 2017, 445,89-96.

(Reply)

              We have discussed and cited some of these papers in the Introduction as follows.

(revised manuscript  p. 1, line 32)

              Semiconductor-based gas sensors are effective for respiratory gas analysis from the standpoints of cost, compactness, and power consumption [6–11]. Su et al. reported that for NO2 sensing, a wearable, alveolus-inspired active membrane sensor based on the WO3 system showed excellent selectivity and thermal and humidity stability [6]. Jha et al. prepared chemiresistive sensors using MoSe2 nanoflakes for ppb-level H2S gas detection [7]. Li et al. synthesized CuO nanoflakes by hydrothermal method and used them to fabricate a sensor, which showed ppb-level H2S gas response [8].

(revised manuscript  p. 8, line 193)

  1. Su, Y.; Wang, J.; Wang, B.; Yang, T.; Yang, B.; Xie, G.; Zhou, Y.; Zhang, S.; Tai, H.; Cai, Z.; Chen, G.; Jiang, Y.; Chen, L.-Q.; Chen, J. Alveolus-Inspired Active Membrane Sensors for Self-Powered Wearable Chemical Sensing and Breath Analysis. ACS Nano 2020, 14, 6067–6075.
  2. Jha, R.K.; D’Costa, J.V.; Sakhuja, N.; Bhat, N. MoSe2 Nanoflakes Based Chemiresistive Sensors for ppb-Level Hydrogen Sulfide Gas Detection. Sens. Actuators B 2019, 297, 126687.
  3. Li, Z.; Wang, N.; Lin, Z.; Wang, J.; Liu, W.; Sun, K.; Fu, Y.Q.; Wang, Z. Room-Temperature High-Performance H2S Sensor Based on Porous CuO Nanosheets Prepared by Hydrothermal Method. ACS Appl. Mater. Interfaces 2016, 8, 20962–20968.
  4. Sarfraz, J.; Fogde, A.; Ihalainen, P.; Peltonen, J. The Performance of Inkjet-Printed Copper Acetate Based Hydrogen Sulfide Gas Sensor on a Flexible Plastic Substrate – Varying Ink Composition and Print Density. Appl. Surf. Sci. 2018, 445, 89–96.

(Over)

Round 2

Reviewer 1 Report

The authors did not addressed the comments satisfactorily and some of the experiments are promised to be waited to carry on. If the author decide to proceed with the experiments, the manuscript can be reconsidered again after the finishing of the experiments.